# Enhancing Annotation Efficiency with Machine Learning: Automated Partitioning of a Lung Ultrasound Dataset by View

**DOI:** 10.3390/diagnostics12102351

**Published:** 2022-09-28

**Authors:** Bennett VanBerlo, Delaney Smith, Jared Tschirhart, Blake VanBerlo, Derek Wu, Alex Ford, Joseph McCauley, Benjamin Wu, Rushil Chaudhary, Chintan Dave, Jordan Ho, Jason Deglint, Brian Li, Robert Arntfield

**Affiliations:** 1Faculty of Engineering, University of Western Ontario, London, ON N6A 5C1, Canada; 2Faculty of Mathematics, University of Waterloo, Waterloo, ON N2L 3G1, Canada; 3Schulich School of Medicine and Dentistry, Western University, London, ON N6A 5C1, Canada; 4Department of Medicine, Western University, London, ON N6A 5C1, Canada; 5Lawson Health Research Institute, London, ON N6C 2R5, Canada; 6Faculty of Engineering, University of Waterloo, Waterloo, ON N2L 3G1, Canada; 7Division of Critical Care Medicine, Western University, London, ON N6A 5C1, Canada; 8Department of Family Medicine, Western University, London, ON N6A 5C1, Canada

**Keywords:** computer vision, machine learning, annotation, labelling, lung ultrasound, medical imaging, deep learning

## Abstract

Background: Annotating large medical imaging datasets is an arduous and expensive task, especially when the datasets in question are not organized according to deep learning goals. Here, we propose a method that exploits the hierarchical organization of annotating tasks to optimize efficiency. Methods: We trained a machine learning model to accurately distinguish between one of two classes of lung ultrasound (LUS) views using 2908 clips from a larger dataset. Partitioning the remaining dataset by view would reduce downstream labelling efforts by enabling annotators to focus on annotating pathological features specific to each view. Results: In a sample view-specific annotation task, we found that automatically partitioning a 780-clip dataset by view saved 42 min of manual annotation time and resulted in 55±6 additional relevant labels per hour. Conclusions: Automatic partitioning of a LUS dataset by view significantly increases annotator efficiency, resulting in higher throughput relevant to the annotating task at hand. The strategy described in this work can be applied to other hierarchical annotation schemes.

## 1. Introduction

Unlike several mainstream computer vision application domains, annotators of medical imaging datasets must possess a sufficient degree of domain expertise to ensure that ground truth is clinically correct. In many cases, labels must be reviewed by clinical experts prior to being officially admitted to a dataset. Given the cost and limited availability of clinical expertise for such tasks, strategies to accurately automate the labelling of medical imaging datasets are desirable.

Lung ultrasound (LUS) is a well described, portable, inexpensive, and accurate point of care technique to assess respiratory disease at the bedside [1,2,3,4,5,6,7], with potential deployment in a wide variety of environments [8,9]. In comparison to the traditional methods used to image the lungs, such as a CT scan or chest X-ray, LUS displays comparable or improved diagnostic accuracy at a reduced cost [4,5]. There are two broadly categorized regions, or views, of the lung that are acquired: parenchymal (anterior and anterolateral chest) and pleural (posterolateral chest) [3,10,11]. Each of these views interrogate different anatomic areas of the lung that may contain separate and distinct disease processes [12]. For example, as seen in Figure 1, if annotating clips for a classifier that identifies A line and B line artifacts [13,14], annotators would be interested in parenchymal views only, since these artifacts are of greatest clinical importance when seen in these views. Conversely, important findings such as the curtain sign, pleural effusion, or consolidation patterns are sought in the pleural views of the lungs [15]. Additional examples of how this is reflected in a hierarchical annotation workflow are shown in Figure 1. The hierarchical nature of LUS interpretation and annotation provides an opportunity to impose high-level structure by partitioning an otherwise unstructured dataset into two clinical and radiographic groups. If the view of every clip in the dataset is known, then the entire dataset can be partitioned by view, and expert annotators need only be provided with clips for which the view is relevant to the annotating task (see Figure 1). LUS is a particularly important modality for optimizing annotation efforts due to the paucity of individuals with sufficient domain expertise to perform LUS annotation [16,17]. Thus, an approach to automated partitioning based on view represents a key opportunity to improve annotation throughput and optimize workforce allocation, while providing a model that is clinically relevant [18,19].

Solutions have been proposed to offset the cost of annotating medical images. Multiple studies have explored the use of active learning, a special case of machine learning where the learner can query a user to label new data points [20]. The direct incorporation of human intervention in the active learning process has been shown to improve both the annotating accuracy and efficiency [21].

The process of leveraging a small, annotated subset of a larger dataset to generate new labels that will be added to a training set has also been explored with notable success. Gu et al. [22] used an annotated training set with 20,000 examples to generate labels for a 100,000-example dataset. This study exhibited a significant improvement in model performance when 80,000 automatically generated labels were added to the human-annotated training set for the purpose of classification [22]. A similar method was used to efficiently label data in [23], where regions of interest in CT examinations were segmented and automatically annotated to circumvent annotation costs.

Deep learning approaches that have been trained for automatic annotating have rivalled the performance of domain experts [24,25]. In this case, radiology reports were used to generate chest X-ray labels. The performance of radiologists was used as a benchmark for model performance, and the margin between the resulting predictions by their deep network and the expert annotator was narrow. This evidence suggests that similar methodologies can be used to rival the annotating accuracy of medical professionals. These findings are encouraging for our work, as the benchmark for this automatic annotating method is also the annotating performance of medical experts. Success in these other domains provide justification for our current work.

The objective of this work is to develop a deep learning solution for automatic LUS view annotation that effectively improves the efficiency of downstream annotation tasks. In particular, a neural network capable of distinguishing parenchymal from pleural LUS views is developed, validated, and used to partition a sample LUS dataset by view. A downstream view-specific annotation task is then performed on both the partitioned dataset and an equally-sized non-partitioned dataset by the same annotation team to investigate whether automatic view annotation improved their efficiency and throughput. We aim for our methods to form the foundation for an improved, more cost-effective LUS annotation workflow that can be applied to other annotation schemes with a hierarchical organization.

## 2. Materials and Methods

### 2.1. Data Curation and Annotation

All data in this study were collected retrospectively from our institutional point-of-care ultrasound database (Qpath E, Port Coquitlam, BC, Canada). To generate ground truth labels, all clips were uploaded to an online platform (Labelbox, San Francisco, CA, USA), where they were annotated by a team of medical professionals trained in LUS. Project oversight, including ambiguous or difficult examples, was provided by an international expert in LUS. Annotation tasks were divided into 200 clip benchmarks for annotators and clip-level classifications were applied, including the view (parenchymal vs. pleural), findings relevant to the respective view (see Figure 1), and quality markers (inappropriate gain, depth, composition, etc.). Annotators also had the option to discard clips that did not meet diagnostic or machine learning standards. Examples include inappropriate ultrasound exams (such as an echocardiogram), user-applied text within the ultrasound image, and removal of the ultrasound probe from the patient’s chest during the video clip. Lastly, annotators had a *skip* option to reserve clips for future annotation. This option was applied when the clip in question did not match the current annotation goals (e.g., a pleural clip when the goal was the annotation of parenchymal findings). The labelling platform automatically tracked the time taken to label or skip clips, which facilitated analysis of annotator efficiency.

### 2.2. View (Parenchymal vs. Pleural) Classifier

#### 2.2.1. Clip-Level Data

To train the neural network, a class-balanced dataset of 2908 LUS clips (1454 parenchymal and 1454 pleural clips) was randomly selected from data previously annotated as described in Section 2.1. By convention, parenchymal and pleural were assigned the negative and positive class, respectively. The details of our training dataset are provided in Table 1.

#### 2.2.2. Frame-Based Data

As the view of an individual LUS image (hereafter referred to as “frame”) can typically be discerned by clinicians, we sought to train a frame-based classification model that could predict the view of a particular LUS frame, where the ground truth view of the frame was the view of the clip (determined by the annotator). Dividing the videos into constituent frames greatly expanded the size of the dataset to 369,832 parenchymal and 330,191 pleural frames. Clip-level predictions could subsequently be inferred from the frame-level predictions using a clip classification algorithm (see Section 2.2.5).

#### 2.2.3. Dataset Pre-Processing

After deconstructing the clips into composite frames, all information external to the ultrasound beam (e.g., vendor logos, depth markers) was removed using ultrasound masking software (AutoMask, WaveBase Inc., Waterloo, ON, Canada). The frames were then resized to 128×128 pixels using bilinear interpolation and fed to the model in RGB channel format. During training, the frame dataset was augmented by applying the following transformations stochastically: random zooming inward/outward by up to 20%, horizontal flipping, brightness shifting by up to 20%, contrast shift of up to 10%, and rotation clockwise/counterclockise by up to $\frac{\pi }{4}$ radians.

#### 2.2.4. Model Architecture

We employed the EfficientNetB0 architecture as the base model [26], with weights pre-trained on ImageNet [27]. The head of the EfficientNetB0 network was replaced with a 2D global average pooling layer, followed by dropout (with dropout rate 0.3), a 128-node fully connected layer with ReLU activation, and a 1-node fully connected output layer with sigmoid activation. The model’s output was the probability *p* that a LUS frame was a pleural view. The predicted frame-level class was taken to be *pleural view* if *p* was at least 0.5 and *parenchymal view* otherwise.

Multiple convolutional neural network architectures were considered for the frame classification task. The weights of each architecture were initialized with pretrained ImageNet [27] weights. A variable number of the first layers in the architecture were kept frozen throughout training. We observed significant overfitting with all architectures studied other than EfficientNetB0: The other architectures achieved an area under the receiver operating curve (AUC) score of at least 0.999 on training data, but consistently obtained significantly lower accuracy on the validation set (see Appendix B). Most of these alternative architectures have more capacity than required for the present task. The EfficientNetB0 architecture, which is more compact, exhibited less overfitting. It was therefore designated as the frame classification architecture. In addition, EfficientNetB0 offers a significant boost in training and inference efficiency compared to other contemporary deep convolutional architectures [26].

#### 2.2.5. Clip Predictions

Since the neural network performed frame-based classification, it was necessary to devise a method to convert a series of outputs into clip-level predictions. Classifying clips in this manner facilitates a direct comparison against our expert annotations and more faithfully resembles clinical, dynamic LUS interpretation. Our approach was based on the clip classification method described in [28]. In summary, the clip prediction was taken to be the positive class if there was at least τ∈N consecutive frames with a prediction probability exceeding the classification threshold t∈[0,1]. Such logic is also applicable to LUS view classification because some frames in pleural clips may resemble parenchymal frames due to the curtain sign artifact (created by movement of aerated lung into and out of view during inspiration and expiration), but not vice versa. To reduce noise in frame-level predictions, we smoothed the frame-level predictions by computing a moving average with a window of width w∈N before applying the existing clip classification method. A visual representation of the hyperparameters involved in generating a clip-level prediction from a series of constituent frame-level predictions is provided in Figure 2.

#### 2.2.6. Validation Strategy

To verify the choice of model architecture and clip prediction hyperparameters, 10-fold cross validation was conducted with the training set. The folds were split by patient ID to prevent data leakage. Values of τ,t,andw were selected via grid search to maximize the average validation set accuracy across all folds. All 14,400 parameter combinations across τ,w∈{1,2,…,40} and t∈{0.1,0.2,…,0.9} were considered in the analysis. We then completed a final training run with a dedicated test split to estimate how well the clip classification method would perform on unseen clips from our database.

To evaluate the clip classification method on unseen data, we sampled a disjoint holdout set of *n* clips from the unannotated LUS database. The holdout set (described in Table 1) was annotated by the standard team as outlined in Section 2.1. To determine the size of the holdout set, we conservatively assumed that the standard annotation team would achieve 96% accuracy on unseen data when compared with the clinical expert’s annotations. Given that we require 95% confidence that the accuracy on the holdout set will lie within ±M of the conservative estimate of *A*, *n* can be calculated using Cochran’s formula for sample size estimation [29].
(1)$$n=\frac{Z_{\alpha }^{2}A\left(1-A\right)}{M^{2}}$$

In the above, $Z_{\alpha }$ is the Z-value corresponding to a α confidence range, and *M* is the margin of error. Applying Equation (Equation 1) with α=95%, M=1.25%, and A=0.960, we obtained n=945 for the size of the holdout set. The accuracy of the clip classification method was compared with that of the standard annotators, where the ground truth was taken as the LUS expert’s decision.

A summary of the complete view classification workflow described in Section 2.2, from pre-processing to classification and analysis, is provided in Figure 3.

### 2.3. Automating the View Annotation Task

#### 2.3.1. Partitioning a LUS Dataset by View

To investigate the utility of the view classifier as an automatic annotation tool, we deployed the model on a distinct set of 2000 clips from the unannotated LUS database and partitioned the data by view prediction. The partitioning criteria was based on the predicted clip-level class as well as the average frame-level prediction probability. In particular, parenchymal-predicted clips with an average frame-level (pleural) prediction probability less than 0.3 were selected to form a parenchymal-specific auto-partitioned dataset. An average frame-level probability of 0.3 was chosen as the threshold for partitioning given the optimal classification threshold (t=0.7) that was observed on the validation set (see Section 3) as well as to minimize the number of pleural clips that would appear in the partitioned dataset. In total, 823 clips met the partitioning criteria, from which 780 were randomly selected for inclusion in the final dataset used for the downstream annotation task. Details of this dataset are available in Table 2.

#### 2.3.2. The Annotation Task

To study the effect of automatic view partitioning on annotator efficiency, a downstream parenchymal-specific annotation task was performed on the aforementioned auto-partitioned dataset. The same annotation task was also performed on a 780-clip, distinct, non-partitioned (control) dataset for comparison (for details, see Table 2). Four experienced members of our annotation team participated in the task, with each member annotating 195 clips from both the control and auto-labelled set as separate annotation tasks (sprints). Sprints were completed in a randomized order, with two members completing the control sprint first, and two completing the auto-partitioned sprint first. Annotators were asked to label all parenchymal and non-usable clips according to the workflow described in Section 2.1, while *skipping* all pleural clips. The effect of automatic view annotation on our overall annotation workflow is outlined in Figure 4.

#### 2.3.3. Statistical Analysis

A one-way paired Student’s t-Test was used to compare control to auto-annotated for each annotation efficiency metric to test for statistical significance. All data are presented as mean ± standard deviation.

## 3. Results

### 3.1. View Classifier Validation

Table 3 details the results of the 10-fold cross validation experiment as well as the performance on the holdout set. The model was evaluated in terms of positive (pleural) and negative (parenchymal) predictive value, given the intended function as an automatic annotation tool: if partitioning for a parenchymal-specific labelling task, then we would aim to minimize the number of false negatives in our dataset (maximize the negative predictive value). If partitioning for a pleural-specific labelling task, then we would aim to minimize the number of false positives in our dataset (maximize the positive predictive value).

#### 3.1.1. Frame-Based Performance

The area under (AUC) the receiver-operator curve (ROC) of our frame-based neural network averaged 0.959(±0.015) on our 10-fold cross validation experiment (Figure 5A) and 0.966 (Figure 5B) on our unseen holdout set. The corresponding frame-level confusion matrices indicated a low proportion of incorrect predictions (Figure 5C,D). This frame-wise performance was deemed satisfactory by clinical team members.

#### 3.1.2. Clip-Based Performance

To evaluate our classifier at the clip-level, an optimal clip classification hyperparameter set was required. The parameter set (τ,t,w)=(7,0.7,17) was found to maximize the average validation set accuracy across each fold for each τ,w∈{1,2,…,40} and t∈{0.1,0.2,…0.9}. The clip-wise performance metrics reported in Table 3 were obtained using this designated parameter set. As shown in Figure 5, the corresponding clip level confusion matrices for both the 10-fold cross-validation experiment (Panel E) and inference on the holdout set (Panel F) showed a high percentage of correct predictions.

Using the results of the holdout set inference, we then sought to estimate how the model would perform if deployed on the remainder of our LUS database as an automatic view annotation tool. By considering the clip-level accuracy obtained on the holdout set (0.925) as a point estimate of our classifier’s performance, we applied Cochran’s formula (Equation (Equation 1)) to estimate that the true accuracy on the remaining unannotated database would lie within a range of 0.925±0.017 with 95% confidence. Therefore, we estimate that the true accuracy, applied to the entire LUS database, is within [0.908,0.942] at the clip level with 95% confidence. The accuracy of our clinical annotation team, as evaluated on the same holdout set, was 0.991.

#### 3.1.3. Frame-Based Explainability

To audit the neural network decisions and instill further confidence in our model at the frame-level, a series of Grad-CAM++ [30] explanations for unseen frames was manually examined by annotators. Annotators largely agreed that the heatmaps highlighted regions considered important for discerning the view of a LUS frame. Figure 6 provides some illustrative examples of correctly and incorrectly classified frames. A post hoc error analysis by clinical team members revealed that false negative predictions were most common for frames where the diaphragm was not visible or obscured. This observation further supports the model’s decision-making ability, given that for a clinician, the diaphragm is a critical structure required for the sonographic landmarking of the pleural view.

#### 3.1.4. Clip-Based Explainability

Although informative, many LUS artifacts cannot be fully captured by static frame-based explainability methods given the dynamic nature of clip acquisition and interpretation. Therefore, to investigate these dynamic artifacts in detail and gain further confidence in our clip-level predictions, we sought to visualize how the predicted frame-level probabilities change over the duration of a given clip. To do so, we generated prediction probability time series plots and overlaid them onto the respective masked LUS clips. A temporal indicator was then added to the graph to create an animation. Illustrative examples of these plots for correctly predicted and incorrectly predicted clips are given in Figure 7, with corresponding animations linked in the figure caption. A clinical post hoc analysis of these animations revealed that our clip-prediction method, in general, is successful in generating accurate clip-level predictions when dynamic artifacts common to LUS interpretation are observed. In particular, the majority of clips displaying the curtain sign artifact are correctly predicted as pleural (Figure 7A; Figure A2A), despite the oscillation in frame-level prediction probability that is observed (and expected). Furthermore, analysis of our incorrectly predicted clips revealed that false positives and negatives were commonly the result of poor acquisition technique. For example, there were several cases where the LUS user moved between parenchymal and pleural views during the same clip (for an example, see Figure A2G). There were also cases where structures indicative of the pleural view, such as the diaphragm or the liver, were either not visualized or were obscured by rib shadowing artifacts (Figure A2F) or aerated lung (Figure A2H).

### 3.2. Automating the View Annotation Task

#### 3.2.1. Performance on an Auto-Partitioned Dataset

Of the 780 clips included in our disjoint dataset auto-partitioned by parenchymal view prediction, 701 were identified as true parenchymal views by our clinical team. Of the 79 clips remaining, 35 were misclassified as pleural views and 44 were discarded for quality control, as described in Section 2.1. Excluding the discarded clips from the analysis, our classifier achieved an accuracy of 701/736=0.952 on this unseen dataset. This is equivalent to the negative predictive value, given that no pleural predictions were included in the dataset. Comparing these results to that of our holdout set and cross validation experiment, we observed a 7.1% and 2.7% improvement in negative predictive value, respectively. This increase in performance is likely the result of our partitioning criteria: By selecting clips with a pleural prediction probability less than 0.3, we reduced the number of false positives appearing in our final partitioned dataset.

#### 3.2.2. Annotation Efficiency

Automatically partitioning by view significantly increased the efficiency of a downstream parenchymal-specific annotation task—the number of relevant (parenchmal) clips included in the 780-clip datasets increased from 351 to 701, while the number of irrelevant (pleural) clips decreased from 383 to 35. The number of clips discarded for quality control was similar (44 in the auto-partitioned dataset and 46 in the control dataset). The lower prevalence of pleural clips in the auto-partitioned dataset (−45%) resulted in significant time savings for annotators, as the average time required to skip a pleural clip was 8.5 s (averaged over the combined 1560-clip dataset). As shown in Figure 8A, the annotators produced more relevant parenchymal labels/hour in the auto-partitioned sprints (176±30) than in the control sprints (121±24;p=0.04). The increase in parenchymal labels/hour corresponded with a decrease in the number of irrelevant pleural clips being skipped per hour (Figure 8B; 131±11 (control) vs. 9±4 (auto-partitioned); p<0.001) and the time spent skipping pleural clips (Figure 8C; 12.6±5.3 min (control) vs. 2.1±0.8 min (auto-partitioned); p=0.02).

## 4. Discussion

In this work, a method capable of distinguishing between parenchymal and pleural LUS views with 92.5% accuracy was developed, validated, and deployed as an automated view annotation tool. The automatic partitioning of a 780-clip LUS dataset by view led to a 42 minute reduction in downstream manual annotation time and resulted in the production of 55±6 extra relevant labels per hour. Our methods form the foundation for an improved annotation workflow that is more efficient, more cost-effective, and applicable to similar hierarchical labelling tasks.

The performance of our clip prediction method on unseen data (displayed in Table 3) was deemed acceptable for internal annotation purposes. Although the accuracy trailed 6.5% behind the clinical annotation team, we demonstrated that implementing an automatic annotation workflow resulted in significant time savings on a sample downstream annotation task. In particular, by not examining the extra 348 irrelevant pleural clips screened out by the view classifier in our sample 780-clip datasets, the annotation team saved 42 min. Extrapolating these results to our remaining unannotated 100,000-clip LUS database, we estimate that automatic view annotation would save the annotation team over 4 days (8.5 s×(0.49−0.04)×100,000clips×1day/86,400 s) when accumulating a dataset of 45,000 parenchymal clips (assuming the same false positive rate of our sample auto-partitioned dataset (0.04), pleural frequency of our sample control dataset (0.49), and average time to skip a pleural clip (8.5 s, see Section 3.2.2)). Expensive expert annotation efforts could then be reallocated to more challenging annotation tasks.

Our approach differs from other automated workflow-enhancing annotation strategies, due to the hierarchical nature of the annotation task at hand. First, we require image-level LUS data only, whereas many others [23,24,25] rely on the presence of additional text data from corresponding clinical reports. Secondly, unlike other methods that seek to minimize the number of annotations required for a specific supervised learning task [20,21], we sought to minimize the time required to annotate a dataset of fixed size with multiple relevant labels downstream in the hierarchy (Figure 1). The resultant annotated dataset is more versatile, since it can be used in the development of multiple classifiers. Further, a similar approach could be taken to automatically partition all parenchymal clips into sets containing either A lines or B lines [28]. Annotation tasks deeper in the hierarchy include lung sliding identification (for clips containing A lines) and B line severity classification (for clips containing B lines).

Despite the aforementioned novelties, the present study is not devoid of limitations. The frame classifier was trained on data from one healthcare institution, hindering application to datasets gathered from external institutions. For most of the downstream annotation tasks, external validation is central to the establishment of model generalizability. In future work, this could be addressed by fine-tuning our classifier on data from external healthcare institutions.

Another future investigation could focus on retraining the frame classifier with an augmented training set that includes automatically annotated LUS clips. Gu et al. [22] witnessed an improvement in model performance using the above procedure. Second, given the comparatively lower metrics for pleural views (likely due to the greater diversity of both radiographic and clinical findings compared to parenchymal views), increasing the proportion of pleural clips in the training set may improve performance.

The classifier developed in this work has utility beyond automatic view annotation. Firstly, it may form the foundation for novel classifiers capable of identifying unique temporal LUS signatures. For example, by visualizing frame prediction probabilities over time, we identified a signature oscillatory pattern that could potentially be used to identify the curtain sign pattern (for examples, see Figure 7A and Figure A2A). In terms of clinical utility, the step-wise deployment of relevant classifiers (view, A line vs. B line, lung sliding, B line severity, etc.) could form the backbone of completely automated LUS interpretation at the bedside. View classification would act as the first step in this hierarchy, ensuring that a potentially novice user has the ultrasound probe in the correct location.

## 5. Conclusions

We describe the development of a deep learning model to accurately partition a large LUS dataset by view. To our knowledge, this is the first description of a method wherein a relatively small subset of a dataset was used to develop a classifier that can automatically partition the rest of an unannotated dataset. Our automated approach considerably improved annotation efficiency, resulting in higher throughput relevant to the annotating task at hand. We propose that this approach can be applied to other unannotated datasets to save considerable manual annotation time and effort. In the clinical environment, view classification could form the backbone of a completely automated LUS interpretation system, where clips are triaged to appropriate classifiers based on the predicted view. Future work involves automatically partitioning the remaining unannotated portion of our LUS database based on other clinical findings downstream in the hierarchy to further optimize annotation resource allocation.

## Figures and Tables

**Figure 1 diagnostics-12-02351-f001:**
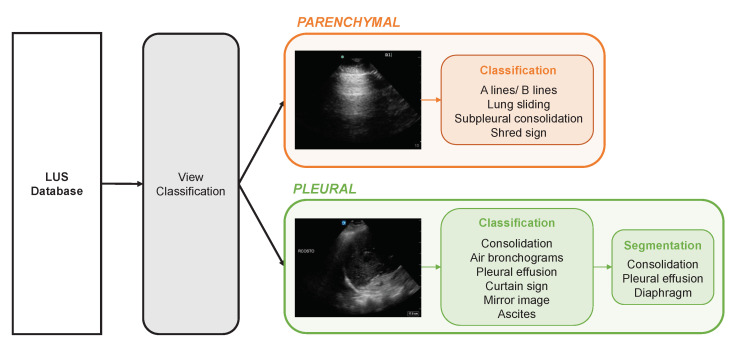
Summary of the hierarchical annotation workflow. LUS classification tasks are view-specific. Automation of the view classification step separates LUS clips. Further, segmentation tasks can subsequently be stratified by classification.

**Figure 2 diagnostics-12-02351-f002:**
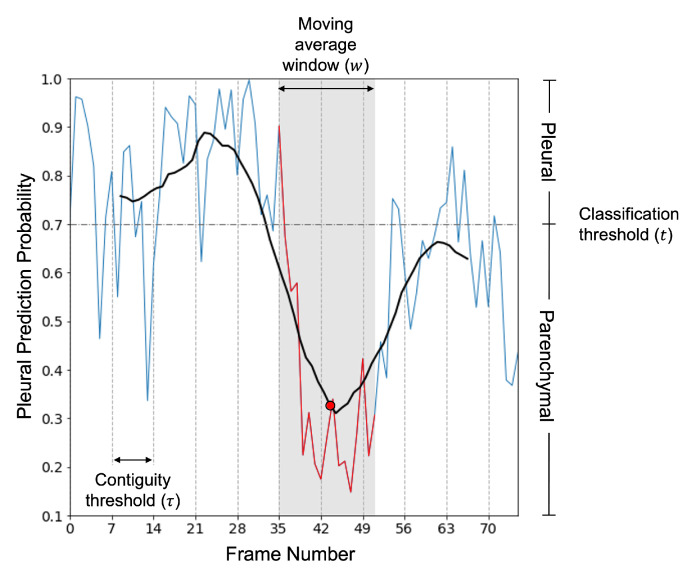
Visual representation of the clip prediction method. For each clip, the raw frame-level prediction probabilities outputted from the neural network (blue curve) are smoothed by computing a moving average (black curve): each point (red dot) on the smoothed prediction curve represents the average of a set of *w* consecutive frame-level prediction probabilities (red curve). The clip is predicted as pleural if τ contiguous smoothed predictions meet or exceed the classification threshold *t*, and parenchymal otherwise. A true positive (pleural) clip is shown, as predicted using the optimal hyperparameter set (τ=7,t=0.7,w=17).

**Figure 3 diagnostics-12-02351-f003:**
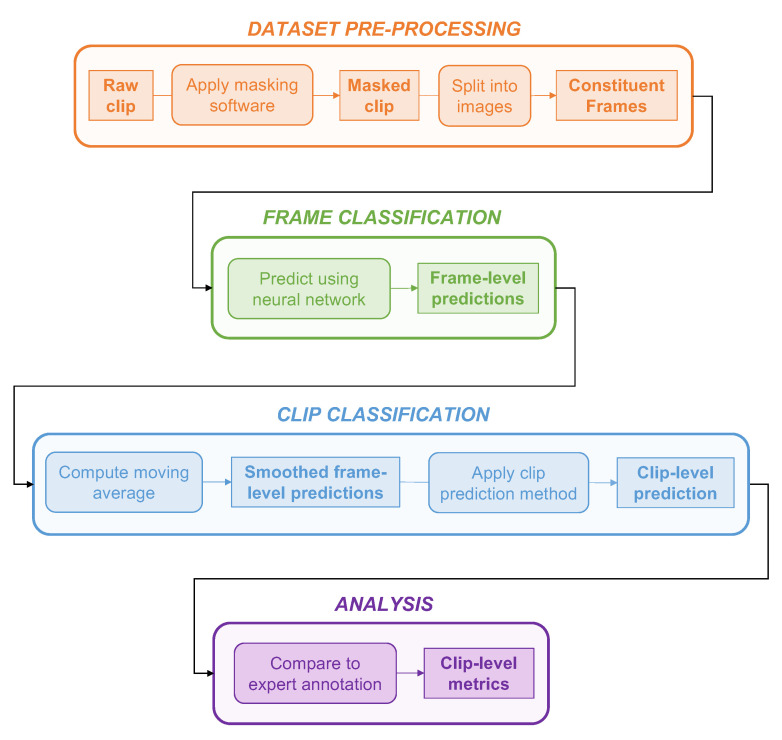
Summary of the view classification workflow. **Data pre-processing:** raw clips were masked to remove information external to the ultrasound beam, then deconstructed into constituent frames. **Frame classification:** processed frames were inputted into the neural network, which predicted the probability that the input frame was a pleural view. **Clip classification:** A moving average was computed over the series of composite frames of a given clip. The smoothed frame-level predictions were then inputted into the contiguous clip prediction method outlined in [28] to generate a whole clip-level prediction. **Analysis:** Clip-level predictions were compared to expert clinical annotations.

**Figure 4 diagnostics-12-02351-f004:**
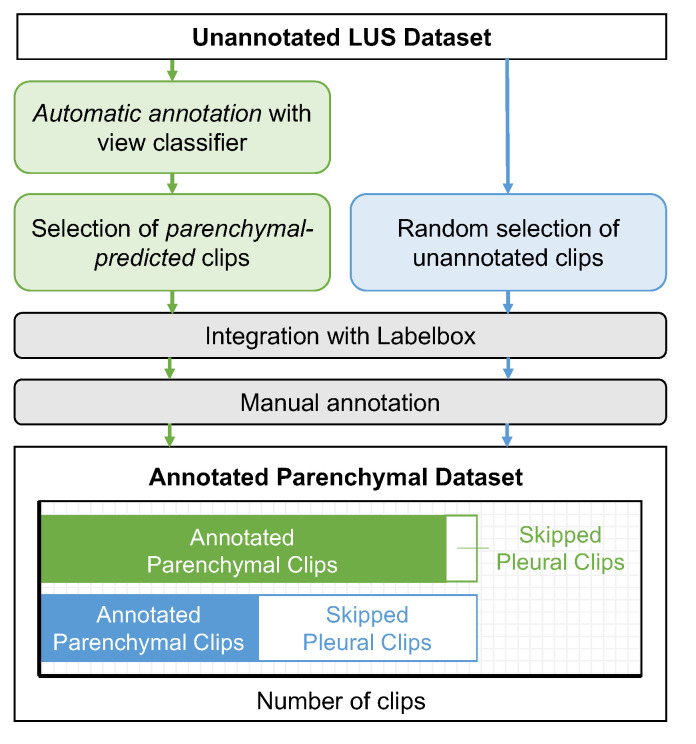
Annotation workflow with (green) and without (blue) automatic view annotation for a parenchymal-specific annotation task. Annotators need only be provided with parenchymal-predicted clips for a parenchymal-specific labelling task, resulting in more annoated parenchymal clips per labelling sprint and fewer skipped pleural clips, saving annotation time.

**Figure 5 diagnostics-12-02351-f005:**
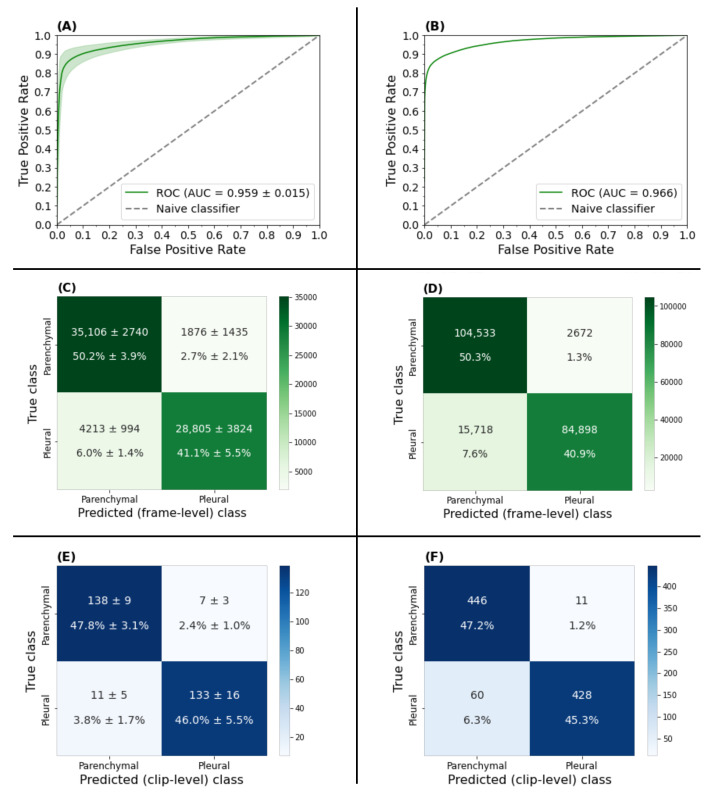
Receiver-operator characteristic curves with corresponding frame (green) and clip-based (blue) confusion matrices for the 10-fold cross validation experiment (**A**,**C**,**E**) and holdout set inference (**B**,**D**,**F**). (**A**) AUC of the 10-fold cross validation experiment averaged 0.959 (±0.015) with the corresponding frame and clip-based confusion matrix results in (**C**) and (**E**), respectively. (**B**) Inference on the holdout set yielded an AUC of 0.966 with the corresponding frame and clip-based confusion matrix results in (**D**) and (**F**), respectively.

**Figure 6 diagnostics-12-02351-f006:**
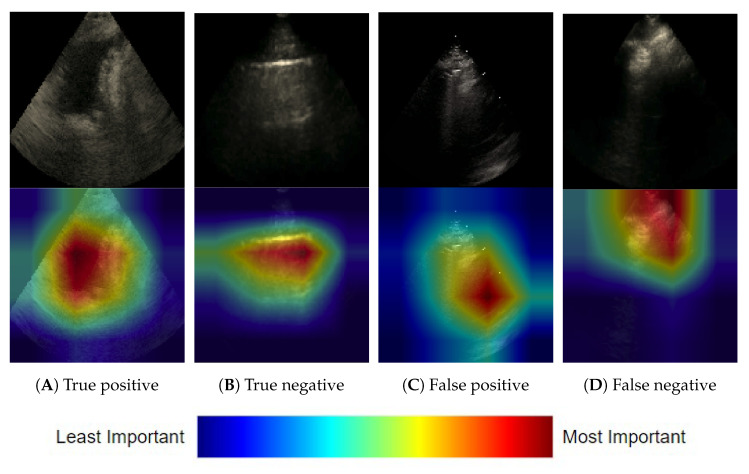
Selected Grad-CAM++ explanations for the neural network model view predictions for single LUS frames. Red regions were the most important to the prediction. The true positive (**A**) is confirmed by the heat map highlighting a pleural effusion which is only seen in the pleural view, and the true negative (**B**) highlights an A line, or reverberation, artifact seen in parenchymals views. The false positive (**C**) highlights the heart likely mistaken as an abdominal organ found in pleural views, and the false negative (**D**) highlights transient parenchymal tissue that comes into frame during inspiration.

**Figure 7 diagnostics-12-02351-f007:**
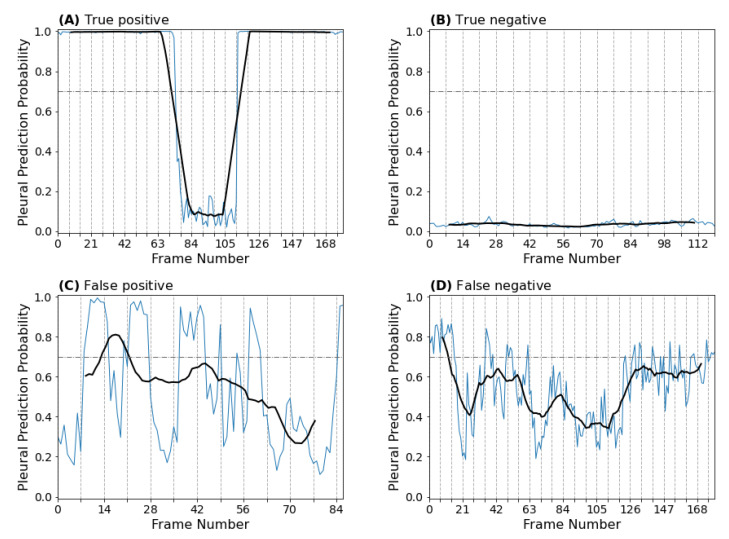
Pleural prediction probability time series for selected true positive (**A**), true negative (**B**), false positive (**C**), and false negative (**D**) clips. The true positive (**A**) clip displays the curtain sign artifact which indicates the lack of pleural pathology (consolidated lung or pleural effusion). The true negative (**B**) clip displays normal lung parenchyma (A line pattern) bordered by rib shadows. The false positive (**C**) clip contains heart tissue, which the model likely mistook for an abdominal organ usually seen in pleural views. The diaphragm is largely missing from the false negative clip (**D**), with only a sliver appearing on a few occasions that correspond to bursts in pleural prediction probability; however, the average probability does not remain above the classification threshold long enough to meet the contiguity threshold. Appendix A.

**Figure 8 diagnostics-12-02351-f008:**
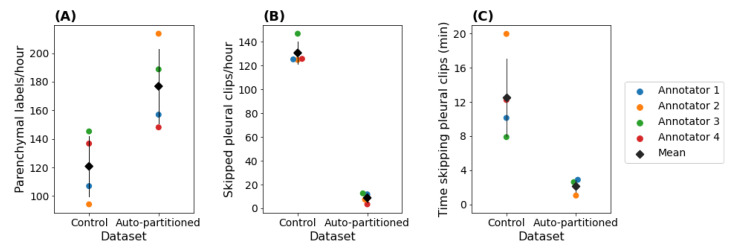
Efficiency analysis of control (non-partitioned) and auto-partitioned (parenchymal-predicted) sprints from the parenchymal-specific labelling task. Time metrics exported from the labelling platform were used to determine the rate of parenchymal labels/hour (**A**), skipped pleural clips/hour (**B**), and the time spent skipping pleural clips (**C**) for each of the four annotators. The diamond represents the mean, and error bars represent standard deviation.

**Table 1 diagnostics-12-02351-t001:** Characteristics of the datasets used for view classifier training and validation.

	Training Data		Holdout Data	
Clip label	Parenchymal	Pleural	Parenchymal	Pleural
Patients	611	342	441	466
Number of clips	1454	1454	457	488
Frames	369,832	330,191	107,205	100,616
Average clips/patient	2.38	4.25	1.04	1.05
Class-patient overlap	303/650		32/875	
Age (std)	64.0 (17.2)	64.5 (16.2)	64.1 (18.0)	64.4 (17.4)
Sex	Female: 238 (39%)	Female: 134 (39%)	Female: 156 (35%)	Female: 205 (44%)
	Male: 347 (57%)	Male: 193 (56%)	Male: 269 (61%)	Male: 235 (50%)
	Unknown: 26 (4%)	Unknown: 15 (4%)	Unknown: 16 (4%)	Unknown: 26 (6%)

**Table 2 diagnostics-12-02351-t002:** Data characteristics of the control and auto-partitioned datasets used for the parenchymal-specific annotation task.

	Control Data		Auto-Partitioned Data	
Clip Label	Parenchymal	Pleural	Parenchymal	Pleural
Patients	339	371	660	34
Number of clips	351	383	701	35
Average clips per patient	1.04	1.03	1.06	1.03
Patient overlap across classes	25/685		5/689	
Mean age (std)	63.7 (18.1)	64.0 (16.1)	64.0 (16.6)	63.7 (18.3)
Sex	Female: 117 (35%)	Female: 156 (42%)	Female: 259 (39%)	Female: 12 (35%)
	Male: 193 (57%)	Male: 201 (54%)	Male: 374 (57%)	Male: 21 (62%)
	Unknown: 29 (9%)	Unknown: 14 (4%)	Unknown: 27 (4%)	Unknown: 1 (3%)

**Table 3 diagnostics-12-02351-t003:** Metrics for a 10-fold cross validation experiment and the holdout set inference.

		Accuracy		Negative Predictive Value		Positive Predictive Value		AUC
**Dataset**	**Fold**	**Frames**	**Clips**	**Frames**	**Clips**	**Frames**	**Clips**	**Frames**
Training	1	0.944	0.966	0.933	0.972	0.957	0.961	0.973
	2	0.930	0.947	0.897	0.938	0.963	0.954	0.969
	3	0.938	0.969	0.910	0.952	0.972	0.986	0.972
	4	0.913	0.935	0.895	0.921	0.940	0.952	0.941
	5	0.907	0.939	0.856	0.908	0.974	0.970	0.963
	6	0.851	0.872	0.885	0.855	0.812	0.893	0.931
	7	0.914	0.939	0.891	0.935	0.947	0.943	0.956
	8	0.922	0.933	0.916	0.951	0.932	0.911	0.971
	9	0.917	0.939	0.883	0.926	0.968	0.952	0.966
	10	0.890	0.919	0.864	0.891	0.920	0.951	0.940
	Mean	0.913	0.936	0.893	0.925	0.935	0.947	0.959
	(STD)	(0.025)	(0.027)	(0.022)	(0.034)	(0.046)	(0.027)	(0.015)
Holdout	−	0.912	0.925	0.869	0.881	0.969	0.975	0.966

## Data Availability

The details of the deep learning model used in this manuscript are available in Appendix C and the implementation can be found at this project’s GitHub repository: https://github.com/deepbreathe-ai/pleural-vs-parenchymal (accessed on 25 July 2022). The patient data itself is not available for open-source sharing at this time but may be able to be made available in the future.

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
