# Peer review of "Enhancing Annotation Efficiency with Machine Learning: Automated Partitioning of a Lung Ultrasound Dataset by View"

_diagnostics, 2022, doi:10.3390/diagnostics12102351_

Round 1
Reviewer 1 Report
1) Abstract: L1-9. Annotating large medical imaging datasets is an arduous and expensive task, especially when the datasets in question are not organized according to deep learning goals. Here, we propose a method that exploits the hierarchical organization of annotating tasks to optimize efficiency. We trained a machine learning model to accurately distinguish between one of two classes of lung ultrasound (LUS) views using 2908 clips from a larger dataset. Partitioning the remaining dataset by view would reduce downstream labelling efforts by enabling annotators to focus on annotating pathological features specific to each view. In a sample view-specific annotation task, we found that automatically partitioning a 780-clip dataset by view saved 42 minutes of manual annotation time and resulted in 55 ± 6 additional relevant labels per hour. The strategy described in this work can be applied to other hierarchical annotation schemes. The abstract is rumbling and difficult to read. Could you please divide the abstract in different sections (e.g. background, aims, results, conclusions, ..).
2) Introduction. L 20-22. Lung ultrasound (LUS) is an inexpensive imaging modality and is a well described point of care technique to assess respiratory disease [1–5], with potential deployment in a wide variety of environments [6,7]. Please, improve this paragraph and add these references:
a- How to do lung ultrasound. Eur Heart J Cardiovasc Imaging. 2022 Mar 22;23(4):447-449. doi: 10.1093/ehjci/jeab241.
b- High-Resolution Computed Tomography and Lung Ultrasound in Patients with Systemic Sclerosis: Which One to Choose? Diagnostics (Basel). 2021 Dec 7;11(12):2293. doi: 10.3390/diagnostics11122293.
3) Figure 1. Summary of the hierarchical annotation workflow. LUS classification tasks are view-specific. Automation of the view classification step separates LUS clips. Further, segmentation tasks can subsequently be stratified by classification. Please, improve the quality of figures.
4) Introduction. L68-70. A downstream view-specific annotation task was then performed on both the partitioned dataset and an equally-sized non-partitioned dataset to investigate the effect of automatic view annotation on annotator efficiency. Could you please improve the description of the aim of the study?
5) 2. Materials and Methods. Could you please underline the statistical evaluations?
6) 4. Discussion L289-291. In this work, a method capable of accurately distinguishing between parenchymal and pleural LUS views was developed, validated, and deployed as an automated view annotation tool. Could you please underline here the most important results of the study?
7) 5. Conclusions L343-349. We describe the development of a deep learning model to accurately partition a large LUS dataset by view. Our automated approach considerably improved annotation efficiency, resulting in higher throughput relevant to the annotating task at hand. We propose that this approach can be applied to other unannotated datasets to save consider- able manual annotation time and effort. Future work involves automatically partitioning the remaining unannotated portion of our LUS database based on other clinical findings downstream in the hierarchy to further optimize annotation resource allocation. Could you please underline the novelty of the paper and the clinical implications?
Reviewer 2 Report
In this paper, Van Berlo et al. introduce a machine learning model to distinguish lung ultrasound imaging datasets, which seems to be effective by saving the following manual annotation time. Here are some questions that need to be addressed:
1) In Figure 6, the color bars should be added for the ultrasound images and heat maps. Figure A1 has the same problem.
2) In “3.2.2. Annotation Efficiency”, the Auto-partitioned group and Control group were compared. The statistical analysis should be clarified, such as P value, SD or SED, and the detailed analysis methods that were employed for calculating the P values.
3) The authors should double check the “References” part. For example, Ref 1: the page number is perhaps missing. Ref 17 also needs to be checked.
